# RECURRENT AUTO-ENCODER MODEL FOR MULTIDIMENSIONAL TIME SERIES REPRESENTATION

## ABSTRACT

Recurrent auto-encoder model can summarise sequential data through an encoder structure into a fixed-length vector and then reconstruct into its original sequential form through the decoder structure. The summarised information can be used to represent time series features. In this paper, we propose relaxing the dimensionality of the decoder output so that it performs partial reconstruction. The fixed-length vector can therefore represent features only in the selected dimensions. In addition, we propose using rolling fixed window approach to generate samples. The change of time series features over time can be summarised as a smooth trajectory path. The fixed-length vectors are further analysed through additional visualisation and unsupervised clustering techniques.

This proposed method can be applied in large-scale industrial processes for sensors signal analysis purpose where clusters of the vector representations can be used to reflect the operating states of selected aspects of the industrial system.

## 1 BACKGROUND

Modern industrial processes are often monitored by a large array of sensors. Machine learning techniques can be used to analyse unbounded streams of sensor signal in an on-line scenario.

This paper illustrates the idea using propietary data collected from a two-stage centrifugal compression train driven by an aeroderivative industrial turbine (RB-211) on a single shaft. It is an isolated large-scale module which belongs to a major natural gas terminal[1]. The purpose of this modular process is to regulate the pressure of natural gas at an elevated, pre-set level.

At the compression system, numerous sensors are attached to various parts of the system to monitor the production process. Vital real-valued measurements like temperature, pressure, rotary speed, vibration... etc., are recorded at different locations [2].

The system can be treated as a multidimensional entity changing through time. Each stream of sensor measurement is basically a set of real values received in a time-ordered fashion. When this concept is extended to a process with $P$ sensors, the dataset can therefore be expressed as a time-ordered multidimensional vector $\{\mathbb{R}_t^P : t \in [1, T]\}$.

The dataset used in this study is unbounded (i.e. continuous streaming) and unlabelled, where the events of interest (e.g. overheating, mechanical failure, blocked oil filters... etc) are not present. The key goal of this study is to find the representation of multiple sensor data in order to identify patterns and anomalies to assist maintenance and diagnostics. We propose a recurrent auto-encoder model which can be used to provide effective vector representation for multidimensional time series data. Further visualisation and clustering techniques can be applied to assist the identification of patterns.

### 1.1 RELATED WORKS

A comprehensive review conducted by (Bagnall et al., 2017) analysed traditional time series clustering algorithms for unidimensional data. Thay have concluded that Dynamic Time Warping (DTW) can be an effective benchmark for unidimensional time series data representation. On the other

---

[1]A simplified process diagram of the compression train can be found in Figure 7 at the appendix.
[2]A list of all sensors is available in the appendix.

hand, there has been many researches done to generalise DTW to multidimensional level (Vlachos et al., 2006; Gillian et al., 2011; ten Holt et al., 2007; Ko et al., 2005; Petitjean et al., 2012; Liu et al., 2009; Wang et al., 2013; Shokoohi-Yekta et al., 2017; Giorgino, 2009). Most of the studies focused on analysing Internet of Things (IoT), wearable sensors and gesture recognition, where the dimensionality of the examined dataset remains relatively low comparing with large-scale industrial applications such as the one we feature in this paper.

In the neural network area, Srivastava et al. (2015) proposed a recurrent auto-encoder model based on LSTM neurons which aims at learning representation of video data. It achieves this by reconstructing sequence of video frames. Their model was able to derive meaningful representations for video clips and the reconstructed outputs demonstrate similarity based on qualitative examination. Another recent paper by D'Avino et al. (2017) also used LSTM-based recurrent auto-encoder model for video data. Sequence of video frames feed into their model so that it learns the intrinsic representation of the video source. Areas of high reconstruction error indicate deviation from the underlying video source and hence can be used as video forgery detection mechanism.

Similarly, audio clips can treated as sequential data. One analysis by Chung et al. (2016) seek to represent variable-length audio data as fixed-length vector using recurrent auto-encoder model. They found that audio segments which sound alike would have vector representations nearby in space.

There are also few other related works in the realm of time series data. For instance, a recent paper by Malhotra et al. (2017) proposed a recurrent auto-encoder model which aims at providing fixed-length representation for bounded univariate time series data. Their model was trained on a plurality of labelled datasets in order to become a generic feature extractor. Dimensionality reduction of the context vectors via t-SNE shows that the ground truth classification can be observed in their model's extracted features. Another study by Hsu (2017) presented a time series compression algorithm using a pair of RNN encoder-decoder structure and an additional auto-encoder to achieve higher compression ratio. Meanwhile, a seperate research by Lee (2017) used an auto-encoder model with database metrics (e.g. CPU usage, number of active sessions... etc) to identify periods of anomaly by setting threshold on the reconstruction error.

## 2 METHODS

A pair of RNN encoder-decoder structure can provide end-to-end mapping between an ordered multidimensional input sequence and its matching output sequence (Sutskever et al., 2014; Cho et al., 2014). Recurrent auto-encoder can be depicted as a special case of the aforementioned model, where input and output sequences are aligned with each other. It can be extended to the area of signal analysis in order to leverage recurrent neurons power to understand complex and time-dependent relationship.

### 2.1 ENCODER-DECODER STRUCTURE

At high level, the RNN encoder reads an input sequence and summarises all information into a fixed-length vector. The decoder then reads the vector and reconstructs the original sequence. Figure 1 below illustrates the model.

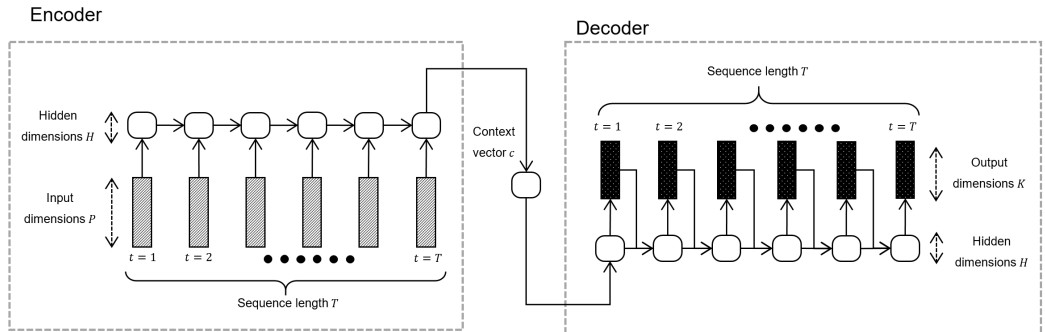

Figure 1: Recurrent auto-encoder model. Both the encoder and decoder are made up of multilayered RNN. Arrows indicate the direction of information flow.

### 2.1.1 ENCODING

The role of the recurrent encoder is to project the multidimensional input sequence into a fixed-length hidden context vector $c$. It reads the input vectors $\{\mathbb{R}_t^P : t \in [1, T]\}$ sequentially from $t = 1, 2, 3, ..., T$. The hidden state of the RNN has $H$ dimensions which updates at every time step based on the current input and hidden state inherited from previous step.

Recurrent neurons arranged in multiple layers are capable of learning complex temporal behaviours. In this proposed model, LSTM neuron with hyperbolic tangent activation is used at all recurrent layers (Hochreiter & Schmidhuber, 1997). An additional improvement of using gated recurrent unit (GRU) neurons (Cho et al., 2014) can also be used but was not experimented within the scope of this study. Once the encoder reads all the input information, the sequence is summarised in a fixed-length vector $c$ which has $H$ hidden dimensions.

For regularisation purpose, dropout can be applied to avoid overfitting. It refers to randomly removing a fraction of neurons during training, which aims at making the network more generalisable (Srivastava et al., 2014). In an RNN setting, Zaremba et al. (2014) suggested that dropout should only be applied non-recurrent connections. This helps the recurrent neurons to retain memory through time while still allowing the non-recurrent connections to benefit from regularisation.

### 2.1.2 DECODING

The decoder is a recurrent network which reconstructs the context vector $c$ back into the original sequence. To exemplify this, the decoder starts by reading the context vector $c$ at $t = 1$. It then decodes the information through the RNN structure and outputs a sequence of vectors $\{\mathbb{R}_t^K : t \in [1, T]\}$ where $K$ denotes the dimensionality of the output sequence.

Recalling one of the fundamental characteristics of an auto-encoder is the ability to reconstruct the input data back into itself via a pair of encoder-decoder structure. This criterion can be slightly relaxed such that $K \leqslant P$, which means the output sequence is only a partial reconstruction of the input sequence.

Recurrent auto-encoder with partial reconstruction:

$$\begin{cases} f_{encoder} : \{\mathbb{R}_t^P : t \in [1, T]\} \to c \\ f_{decoder} : c \to \{\mathbb{R}_t^K : t \in [1, T]\} \end{cases} \quad K \leqslant P \tag{1}$$

In the large-scale industrial system use case, all streams of sensor measurements are included in the input dimensions while only a subset of sensors is included in the output dimensions. This means the entire system is visible to the encoder, but the decoder only needs to perform partial reconstruction of it. End-to-end training of the relaxed auto-encoder implies that the context vector would summarise the input sequence while still being conditioned on the output sequence. Given that activation of the context vector is conditional on the decoder output, this approach allows the encoder to capture lead variables across the entire process as long as they are relevant to the selected output dimensions.

It is important that we recognise that reconstructing part of the data is clearly an easier task to perform than fully-reconstructing the entire original sequence. However, partial reconstruction has practical significance for the industrial application aspect. In real-life scenarios, multiple context vectors can be generated from different recurrent auto-encoder models using identical sensors in the encoder input but different subset of sensors in the decoder output. The selected subsets of sensors can reflect the underlying states of different parts of the system. As a result, these context vectors produced from the same time segment can be used as different diagnostic measurements in industrial context. We will illustrate this in the results section by highlighting two examples.

## 2.2 SAMPLING

For a training dataset of $T'$ time steps, samples can be generated where $T < T'$. We can begin at $t = 1$ and draw a sample of length $T$. This process continues recursively by shifting one time step until it reaches the end of training dataset. For a subset sequence of length $T$, this method allows $T' - T$ samples to be generated. Besides, it can also generate samples from an unbounded time series in an on-line scenrio, which are essential for time-critical applications like sensor data analysis.

---

**Algorithm 1:** Drawing samples consecutively from the original dataset

**Input:** Dataset length $T'$
**Input:** Sample length $T$
**1** $i \leftarrow 0$ ;
**2** **while** $i \leqslant i + T$ **do**
**3** $\quad$ Generate sample sequence $(i, i + T]$ from the dataset;
**4** $\quad$ $i \leftarrow i + 1$;
**5** **end**

---

Given that sample sequences are recursively generated by shifting the window by one time step, successively-generated sequences are highly correlated with each other. As we have discussed previously, the RNN encoder structure compresses sequential data into a fixed-length vector representation. This means that when consecutively-drawn sequences are fed through the encoder structure, the resulting activation at $c$ would also be highly correlated. As a result, consecutive context vectors can join up to form a smooth trajectory in space.

Context vectors in the same neighbourhood have similar activation therefore they must have similar underlying states. Contrarily, context vectors located in distant neighbourhoods would have different underlying states. These context vectors can be visualised in lower dimensions via dimensionality reduction techniques such as principal component analysis (PCA).

Furthermore, additional unsupervised clustering algorithms can be applied to the context vectors. Each context vector can be assigned to a cluster $C_j$ where $J$ is the total number of clusters. Once all the context vectors are labelled with their corresponding clusters, supervised classification algorithms can be used to learn the relationship between them using the training set. For instance, support vector machine (SVM) classifier with $J$ classes can be used. The trained classifier can then be applied to the context vectors in the held-out validation set for cluster assignment. It can also be applied to context vectors generated from unbounded time series in an on-line setting. Change in cluster assignment among successive context vectors indicate change in the underlying state.

## 3 RESULTS

Training samples were drawn from the dataset using windowing approach with fixed sequence length. In our example, the large-scale industrial system has $158$ sensors which means the recurrent auto-encoder's input dimension has $P = 158$. Observations are taken at 5 minutes granularity and the total duration of each sequence was set at 3 hours. This means that the model's sequence has fixed length $T = 36$, while samples were drawn from the dataset with total length $T' = 2724$. The dataset was scaled into $z$-scores, thus ensuring zero-centred data which facilitates gradient-based training.

The recurrent auto-encoder model has three layers in the RNN encoder structure and another three layers in the corresponding RNN decoder. There are $400$ neurons in each layer. The auto-encoder

model structure can be summarised as: RNN encoder(400 neurons/3 layers LSTM/hyperbolic tangent) - Context layer (400 neurons/Dense/linear activation) - RNN decoder(400 neurons/3 layers LSTM/hyperbolic tangent). Adam optimiser (Kingma & Ba, 2014) with $0.4$ dropout rate was used for model training.

### 3.1 OUTPUT DIMENSIONITY

As we discussed earlier, the RNN decoder's output dimension can be relaxed for partial reconstruction. The output dimensionality was set at $K = 6$ which is comprised of a selected set of sensors relating to key pressure measurements (e.g. suction and discharge pressures of the compressor device).

We have experimented three scenarios where the first two have complete dimensionality $P = 158; K = 158$ and $P = 6; K = 6$ while the remaining scenario has relaxed dimensionality $P = 158; K = 6$. The training and validation MSEs of these models are visualised in figure 2 below.

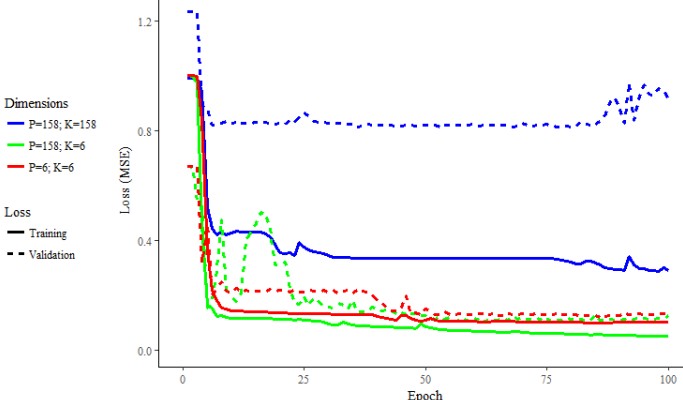

Figure 2: Effects of relaxing dimensionality of the output sequence on the training and validation MSE losses. They contain same number of layers in the RNN encoder and decoder respectively. All hidden layers contain same number of LSTM neurons with hyperbolic tangent activation.

The first model with complete dimensionality ($P = 158; K = 158$) has visibility of all dimensions in both the encoder and decoder structures. Yet, both the training and validation MSEs are high as the model struggles to compress-decompress the high dimensional time series data.

For the complete dimensionality model with $P = 6; K = 6$, the model has limited visibility to the system as only the selected dimensions were included. Despite the context layer summarises information specific to the selected dimensionality in this case, lead variables in the original dimensions have been excluded. This prevents the model from learning any dependent behaviours among all available information.

On the other hand, the model with partial reconstruction ($P = 158; K = 6$) demonstrate substantially lower training and validation MSEs. As all information is available to the model via the RNN encoder, it captures all relevant information such as lead variables across the entire system.

Randomly selected samples in the held-out validation set were fed to this model and the predictions can be qualitatively examined in details. In figure 3 below, all the selected specimens demonstrate similarity between the original label and the reconstructed output. The recurrent auto-encoder model captures the shift mean level as well as temporal variations of all the output dimensions.

### 3.2 CONTEXT VECTOR

Once the recurrent auto-encoder model is successfully trained, samples can be fed to the model and the corresponding context vectors can be extracted for detailed inspection. As we discussed earlier, successive context vectors have similar activation as they are only shifted by one time step. By calculating the correlation matrix of all context vectors and visualising them on a heatmap as in

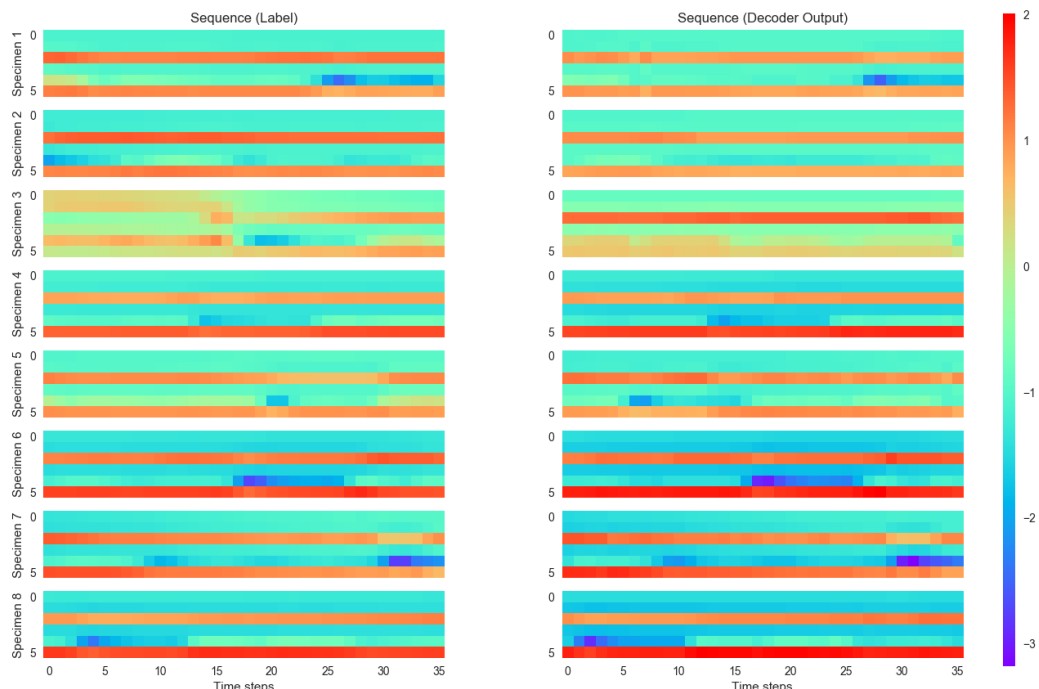

Figure 3: A heatmap showing eight randomly selected output sequences in the held-out validation set. Colour represents magnitude of sensor measurements in normalised scale.

figure 4, it is found that the narrow band around the diagonal has consistently higher correlation. This indicates that successive context vectors are highly correlated.

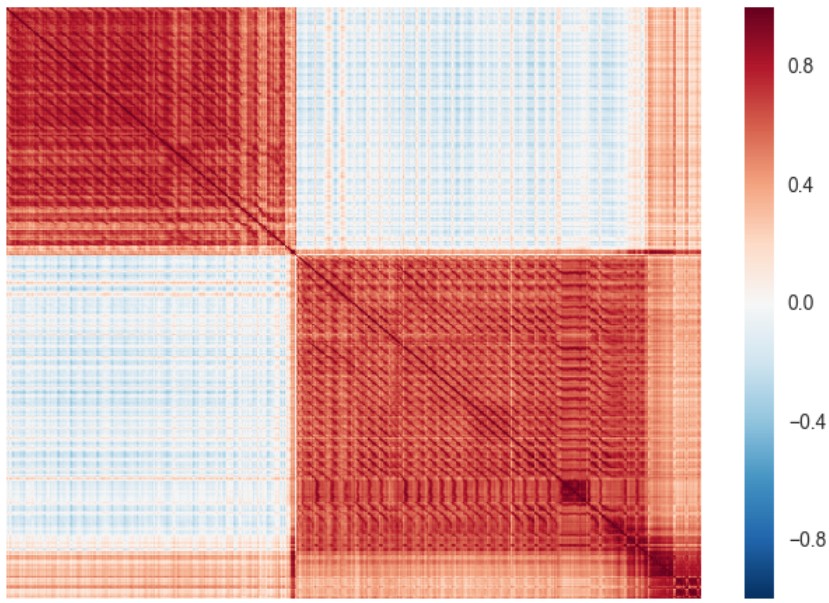

Figure 4: A correlation matrix showing the pairwise correlation of all context vectors. Notice the narrow band around the diagonal always has stronger correlation.

In the model we selected, the context vector $c$ is a multi-dimensional real vector $\mathbb{R}^{400}$. Since the model has input dimensions $P = 158$ and sequence length $T = 36$, the model has achieved com-

pression ratio $\frac{158 \times 36}{400} = 14.22$. Dimensionality reduction of the context vectors through principal component analysis (PCA) shows that context vectors can be efficiently embedded in lower dimensions (e.g. two-dimensional space).

At low-dimensional space, we can use supervised classification algorithms to learn the relationship between vectors representations and cluster assignment. The trained classification model can then be applied to the validation set to assign clusters for unseen data. In our experiment, a SVM classifier with radial basis function (RBF) kernel ($\gamma = 4$) was used. The results are shown in figure 5 below.

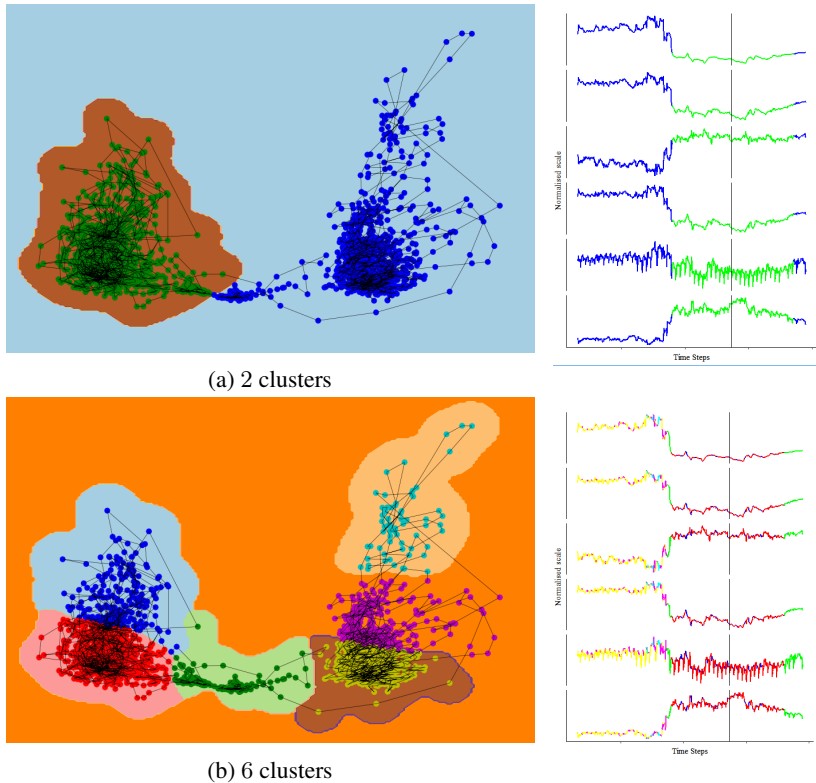

(a) 2 clusters

(b) 6 clusters

Figure 5: The first example. On the left, the context vectors were projected into two-dimensional space using PCA. The black solid line on the left joins all consecutive context vectors together as a trajectory. Different number of clusters were identified using simple $K$-means algorithm. Cluster assignment and the SVM decision boundaries are coloured in the charts. On the right, output dimensions are visualised on a shared time axis. The black solid line demarcates the training set (70%) and validation sets (30%). The line segments are colour-coded to match the corresponding clusters.

In the two-dimensional space, the context vectors separate into two clearly identifiable neighbourhoods. These two distinct neighbourhoods correspond to the shift in mean values across all output dimensions. When $K$-means clustering algorithm is applied, it captures these two neighbourhoods as two clusters in the first scenario.

When the number of clusters increases, they begin to capture more subtleties. In the six clusters scenario, successive context vectors oscillate back and forth between close clusters which correspond to interlacing troughs and crests in the output dimensions. Similar pattern can also be observed in the validation set, which indicates that the knowledge learned by the auto-encoder model is generalisable to unseen data.

Furthermore, we have repeated the same experiment again with a different configuration ($K = 158; P = 2$) to reassure that the proposed approach can provide robust representations of the data. The sensor measurements are drawn from an identical time period and only the output dimensionality $K$ is changed (The newly selected set of sensors is comprised of a different measurements of discharge gas pressure at the compressor unit). Through changing the output dimensionality $K$, this allows us to illustrate the effects of partial reconstruction with different output dimensions. As

seen in figure 6, the context vectors form a smooth trajectory in the low-dimensional space. Similar sequences yield context vectors which are located in a shared neighbourhood. Nevertheless, the clusters found by $K$-means method in this secondary example also manage to identify neighbourhoods of similar sensor patterns.

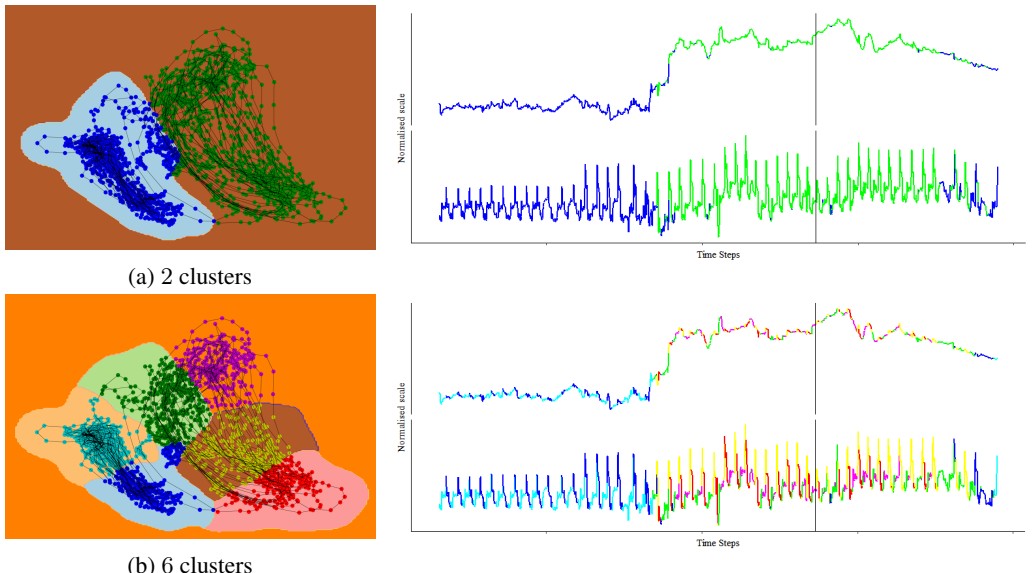

(a) 2 clusters

(b) 6 clusters

Figure 6: The second example. The sensor data is drawn from the same time period as the previous example, only the output dimension has been changed to $K = 2$ where another set of gas pressure sensors were selected.

# 4    DISCUSSION

Successive context vectors generated by windowing approach are always highly correlated, thus form a smooth trajectory in high-dimensional space. Additional dimensionality reduction techniques can be applied visualise the change of features as a smooth trajectory. One of the key contributions of this study is that, the context vectors form distinct neighbourhoods which can be identified through unsupervised clustering algorithms such as $K$-means. The clusters can be optionally labelled manually to identify operating state (e.g. healthy vs. faulty). Alarm can be triggered when the context vector travels beyond the boundary of a predefined neighbourhood. Moreover, this enables us to find the clusters of unlabelled time series data. Clusters of the vector representation can be used by operators and engineers to aid diagnostics and maintenance.

Another contribution of this study is that dimensionality of the output sequence can be relaxed, thus allowing the recurrent auto-encoder to perform partial reconstruction. Although it is clearly easier for the model to partially reconstruct the original sequence, such simple improvement allows users to define different sets of sensors of particular interest. By limiting the number of sensors to include in the output dimension, the context vector can be used to reflect the underlying states of specific aspects of the large-scale industrial process. This ultimately generates more actionable insights and enables users to diagnose the induatrial system. We have demonstrated the use of partial reconstruction by through two examples which graphically show the effects of it.

This proposed method performs multidimensional time series clustering, which can natively scale up to very high dimensionality as it is based on recurrent auto-encoder model. We have applied the method to an industrial sensor dataset with $P = 158$ and empirically show that it can summarise multidimensional time series data effectively.

The model can be generalised to any multi-sensor multi-state processes for operating state recognition. We also recognise that the cost of collecting labelled time series data can be very expensive. This study established that recurrent auto-encoder model can be used to analyse unlabelled and un-

bounded time series data. This opens up further possibilities for analysing IoT and industrial sensors data given that these domains are predominately overwhelmed with unbounded and unlabelled time series data.

Nevertheless, the proposed approach has not included any categorical sensor measurements (e.g. open/closed, tripped/healthy, start/stop... etc). Future research can focus on incorporating categorical measurements alongside real-valued measurements.

DISCLOSURE

The technical method described in this paper is the subject of British patent application GB1717651.2.

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

APPENDIX A

The rotary components are driven by industrial RB-211 jet turbine on a single shaft through a gearbox. Incoming natural gas passes through the low pressure (LP) stage first which brings it to an intermediate pressure level, it then passes through the high pressure (HP) stage and reaches the pre-set desired pressure level. The purpose of the suction scrubber is to remove any remaining condensate from the gas prior to feeding through the centrifugal compressors. Once the hot compressed gas is discharged from the compressor, its temperature is lowered via the intercoolers.

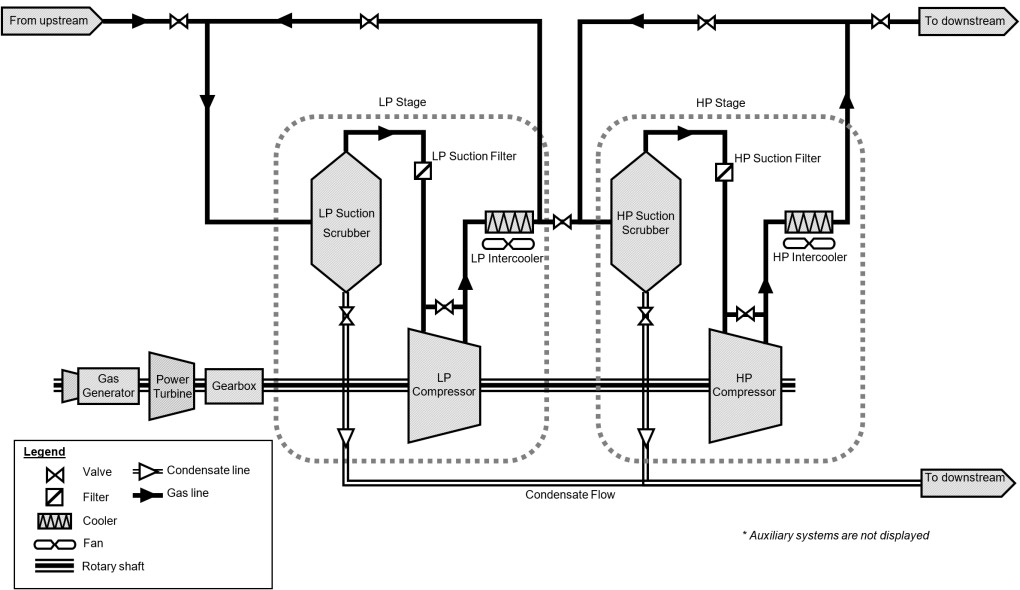

Figure 7: A simplified process diagram of the two-stage centrifugal compression train which is located at a natural gas terminal.

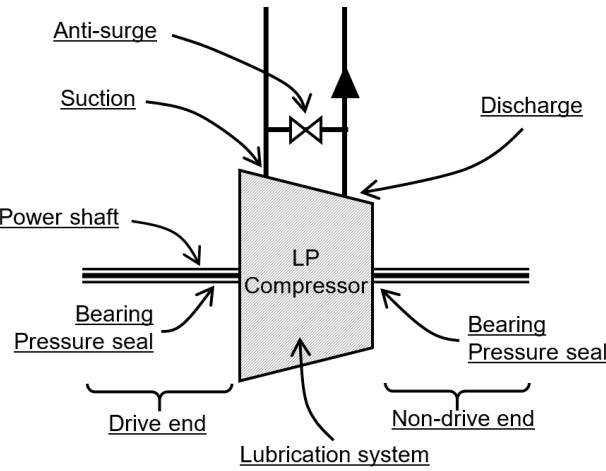

Figure 8: Locations of key components around the centrifugal compressor.

## APPENDIX B

The sensor measurements used in the analysis are listed below:

1. GASCOMPCARBONDIOXIDEMEAS
2. GASCOMPMETHANEMEAS
3. GASCOMPNITROGENMEAS
4. GASPROPMOLWTMEAS
5. PRESSAMBIENT
6. GB_SPEEDINPUT
7. GB_SPEEDOUTPUT
8. GB_TEMPINPUTBRGDRIVEEND
9. GB_TEMPINPUTBRGNONDRIVEEND
10. GB_TEMPINPUTBRGTHRUSTINBOARD
11. GB_TEMPINPUTBRGTHRUSTOUTBRD
12. GB_TEMPLUBOIL
13. GB_TEMPLUBOILTANK
14. GB_TEMPOUTPUTBRGDRIVEEND
15. GB_TEMPOUTPUTBRGNONDRIVEEND
16. GB_VIBBRGCASINGVEL
17. GB_VIBINPUTAXIALDISP
18. GB_VIBINPUTDRIVEEND
19. GB_VIBINPUTNONDRIVEEND
20. GB_VIBOUTPUTDRIVEEND
21. GB_VIBOUTPUTNONDRIVEEND
22. GG_FLOWFUEL
23. GG_FLOWWATERINJECTION
24. GG_FLOWWATERINJSETPOINT
25. GG_POWERSHAFT
26. GG_PRESSAIRINLET
27. GG_PRESSCOMPDEL
28. GG_PRESSCOMPDELHP
29. GG_PRESSCOMPDELIP
30. GG_PRESSDIFBRGLUBOIL
31. GG_PRESSDIFINLETFILTER
32. GG_PRESSDIFINLETFLARE
33. GG_PRESSDIFVALVEWATERINJCTRL
34. GG_PRESSDISCHWATERINJPUMP1
35. GG_PRESSDISCHWATERINJPUMP2
36. GG_PRESSEXH
37. GG_PRESSFUELGAS
38. GG_PRESSHYDOILDEL
39. GG_PRESSLUBEOILHEADER
40. GG_PRESSLUBOIL
41. GG_PRESSMANIFOLDWATERINJ
42. GG_PRESSSUCTWATERINJPUMP
43. GG_SPEEDHP
44. GG_SPEEDIP
45. GG_TEMPAIRINLET
46. GG_TEMPCOMPDEL
47. GG_TEMPCOMPDELHP
48. GG_TEMPCOMPDELIP
49. GG_TEMPEXH
50. GG_TEMPEXHTC1
51. GG_TEMPEXHTC2
52. GG_TEMPEXHTC3
53. GG_TEMPEXHTC4
54. GG_TEMPEXHTC5
55. GG_TEMPEXHTC6
56. GG_TEMPEXHTC7
57. GG_TEMPEXHTC8
58. GG_TEMPFUELGAS
59. GG_TEMPFUELGASG1
60. GG_TEMPFUELGASLINE
61. GG_TEMPHSOILCOOLANTRETURN
62. GG_TEMPHSOILMAINRETURN
63. GG_TEMPLUBOIL
64. GG_TEMPLUBOILTANK
65. GG_TEMPPURGEMUFF
66. GG_TEMPWATERINJSUPPLY
67. GG_VALVEWATERINJECTCONTROL
68. GG_VANEINLETGUIDEANGLE
69. GG_VANEINLETGUIDEANGLE1
70. GG_VANEINLETGUIDEANGLE2
71. GG_VIBCENTREBRG
72. GG_VIBFRONTBRG
73. GG_VIBREARBRG
74. HP_HEADANTISURGE
75. HP_POWERSHAFT
76. HP_PRESSCLEANGAS
77. HP_PRESSDIFANTISURGE
78. HP_PRESSDIFSUCTSTRAINER
79. HP_PRESSDISCH
80. HP_PRESSSEALDRYGAS
81. HP_PRESSSEALLEAKPRIMARYDE1
82. HP_PRESSSEALLEAKPRIMARYDE2
83. HP_PRESSSEALLEAKPRIMARYNDE1
84. HP_PRESSSEALLEAKPRIMARYNDE2
85. HP_PRESSSUCT1
86. HP_PRESSSUCT2
87. HP_SPEED
88. HP_TEMPBRGDRIVEEND
89. HP_TEMPBRGNONDRIVEEND
90. HP_TEMPBRGTHRUSTINBOARD
91. HP_TEMPBRGTHRUSTOUTBOARD
92. HP_TEMPDISCH1
93. HP_TEMPDISCH2
94. HP_TEMPLUBOIL
95. HP_TEMPLUBOILTANK
96. HP_TEMPSUCT1
97. HP_VIBAXIALDISP1
98. HP_VIBAXIALDISP2
99. HP_VIBDRIVEEND
100. HP_VIBDRIVEENDX
101. HP_VIBDRIVEENDY
102. HP_VIBNONDRIVEEND
103. HP_VIBNONDRIVEENDX
104. HP_VIBNONDRIVEENDY

105. HP_VOLDISCH
106. HP_VOLRATIO
107. HP_VOLSUCT
108. LP_HEADANTISURGE
109. LP_POWERSHAFT
110. LP_PRESSCLEANGAS
111. LP_PRESSDIFANTISURGE
112. LP_PRESSDIFSUCTSTRAINER
113. LP_PRESSDISCH
114. LP_PRESSSEALDRYGAS
115. LP_PRESSSEALLEAKPRIMARYDE1
116. LP_PRESSSEALLEAKPRIMARYDE2
117. LP_PRESSSEALLEAKPRIMARYNDE1
118. LP_PRESSSEALLEAKPRIMARYNDE2
119. LP_PRESSSUCT1
120. LP_PRESSSUCT2
121. LP_SPEED
122. LP_TEMPBRGDRIVEEND
123. LP_TEMPBRGNONDRIVEEND
124. LP_TEMPBRGTHRUSTINBOARD
125. LP_TEMPBRGTHRUSTOUTBOARD
126. LP_TEMPDISCH1
127. LP_TEMPDISCH2
128. LP_TEMPLUBOIL
129. LP_TEMPLUBOILTANK
130. LP_TEMPSUCT1
131. LP_VIBAXIALDISP1

132. LP_VIBAXIALDISP2
133. LP_VIBDRIVEEND
134. LP_VIBDRIVEENDX
135. LP_VIBDRIVEENDY
136. LP_VIBNONDRIVEEND
137. LP_VIBNONDRIVEENDX
138. LP_VIBNONDRIVEENDY
139. LP_VOLDISCH
140. LP_VOLRATIO
141. LP_VOLSUCT
142. PT_POWERSHAFT
143. PT_SPEED
144. PT_TEMPBRGDRIVEEND
145. PT_TEMPBRGNONDRIVEEND
146. PT_TEMPBRGTHRUST1
147. PT_TEMPBRGTHRUST3
148. PT_TEMPCOOLINGAIR1
149. PT_TEMPCOOLINGAIR2
150. PT_TEMPEXH
151. PT_TEMPLUBOIL
152. PT_TEMPLUBOILPTSUMP
153. PT_TEMPLUBOILTANK
154. PT_VIBAXIALDISP1
155. PT_VIBAXIALDISP2
156. PT_VIBBRGCASINGVEL
157. PT_VIBDRIVEEND
158. PT_VIBNONDRIVEEND

