# OpenReview forum: "Recurrent Auto-Encoder Model for Multidimensional Time Series Representation"
_ICLR.cc/2018/Conference — Reject_

### Official Review · AnonReviewer1 · 2017-11-19
**substandard quality**

**Rating:** 2
**Confidence:** 4

**Review:**

This writeup describes an application of recurrent autoencoder to analysis of multidimensional time series. The quality of writing, experimentation and scholarship is clearly below than what is expected from a scientific article. The method is explained in a very unclear way, there is no mention of any related work. I would encourage the authors to take a look at other ICLR submissions and see how rigorously written they are, how they position the reported research among comparable works.

---

> ### Author Response · Authors · 2018-01-04
> **Paper updated**
>
> Thanks for your review.
>
> We've added the description of the dataset to the updated paper (e.g. graphs, sensor names, locations, etc). Also added a more detailed description of the model.

---

### Official Review · AnonReviewer2 · 2017-11-27
**Trivial results and obscure data set.**

**Rating:** 2
**Confidence:** 5

**Review:**

The paper describes a sequence to sequence auto-encoder model which is used to learn sequence representations. The authors show that for their application, better performance is obtained when the network is only trained to reconstruct a subset of the data measurements. The paper also presents some visualizations the similarity structure of the learned representations and proposes a window-based method for processing the data.

According to the paper, the experiments are done using a data set which is obtained from measurements of an industrial production process. Figure 2 indicates that reconstructing fewer dimensions of this dataset leads to lower MSE scores. I don’t see how this is showing anything besides the obvious fact that reconstructing fewer dimensions is an easier task than reconstructing all of them.  The only conclusions I can draw from the visual analysis is that the context vectors are more similar to each other when they are obtained from time steps in the data stream which are close to each other. Since the paper doesn’t describe much about the privately owned data at all, there is no possibility to replicate the work. The paper doesn’t frame the work in prior research at all and the six papers it cites are only referred to in the context of describing the architecture.

I found it very hard to distil what the main contribution of this work was according to the paper. There were also not many details about the precise architecture used. It is implied that GRU networks and were used but the text doesn’t actually state this explicitly. By saying so little about the data that was used, it was also not clear what the temporal correlations of the context vectors are supposed to tell us.

The paper describes how existing methods are applied to a specific data set. The benefit of only reconstructing a subset of the input dimensions seems very data specific to me and I find it hard to consider this a novel idea by itself. Presenting sequential data in a windowed format is a standard procedure and not a new idea either. All in all I don't think that the paper presents any new ideas or interesting results.

Pros:
* The visualizations look nice.

Cons:
* It is not clear what the main contribution is.
* Very little information about the data.
* No clear experiments from which conclusions can be drawn.
* No new ideas.
* Not well rooted in prior work.

---

> ### Author Response · Authors · 2018-01-04
> **Paper updated**
>
> Thanks for your review.
>
> We've added a detailed description of the dataset in the updated paper. It is sourced from a large compressor unit and the names of all sensors are also attached in the appendix.
>
> For the related works, we've expanded this section a lot. Traditional methods based on DTW (non-NN) have been cited in the paper. Also more recently published NN-based models such as those auto-encoder applications relating to time series, video and audio have been added too.  We have not found any RNN auto-encoder research relating to large-scale industrial sensor like the one we showcase. Closest are the ones relating to wearable sensors (accelerometers + gyroscope, also cited in the updated paper) but the number of dimensions involved is much smaller.
>
> The benefit of partial reconstruction is not mathematical but purely practical (I also explained this for another reviewer). That reconstruction a selected subset of sensors allows the vector representation to reflect the underlying states of various aspects of the industrial system. Practically, engineers and operators don't need to know whether a machine has failed. Instead, they need to know which part of the machine failed and what kind of failure mode occured (clusters can be found in PCA space of the context vector, which reflects different operating states). This is useful for operators because the raw data is completely unlabelled, by using the proposed approach they can identify the cluster of the current context vector in an on-line setting. Then they can further deduce the operating states of the machine which helps with diasnostic/maintenance.
>
> The model uses encoder/decoder with three layers. LSTM neurons were used and briefly mentioned in a small figure in the original paper but we've now adopted your comment and stated it much clearer in the main text.

---

### Official Review · AnonReviewer3 · 2017-11-28
**Important background information/related work missing**

**Rating:** 4
**Confidence:** 4

**Review:**

This paper proposes a strategy that is inspired by the recurrent auto-encoder model, such that clustering of multidimensional time series data can be performed based on the context vectors generated by the encoding process. Unfortunately, the paper in its current form is a bit thin on content.

Main issues:
No related works (such as those using RNN for time series analysis or clustering of time series data streams etc.) were described by the paper, no baselines were used in the comparison evaluations, and no settings/details were provided in the experiment section. As a result, it is quite difficult to judge the merits and novelty of the paper.

Other issues:
some contribution claims highlighted in the Discussion Section, i.e., Section 4, are arguable and should be further extended. For example, the authors claim that the proposed LSTM-based autoencoder networks can be natively scaled up to data with very high dimensionality. I would like the authors to explain it in more details or empirically demonstrate that, since a LSTM-based model could be computationally expensive. As another example, the authors claim that reducing the dimensionality of the output sequence is one of the main contributions of the paper. In this sense, further elaborations from that perspective would be very beneficial since some networks already employ such a mechanism.

In short, the paper in its current form does not provide sufficient details for the reviewer to judge its merits and contributions.

---

> ### Author Response · Authors · 2018-01-04
> **Paper updated**
>
> Thanks for your review.
>
> A more detailed description of the problem and dataset was added (both in-text and appendix) to the updated paper. We source the sensor data from an large-scale industrial compressor situated at a natural gas terminal. Also a summary diagram was added to provide better understanding of the problem.
>
> Secondly, we've added related works (both NN-based algorithms and non-NN works in this area too). We also found that most of the previous related works were about wearable sensors, where the number of sensor measurements are relatively low comparing with the use case we present in this paper. One contribution of the study is that we applied recurrent aut-encoder model on multidimensional time series where the dimensionity is quite high (hundreds on sensors at a large-scale industrial gas compressor) and empirically demonstrate that the vector representation can effectively summarise multidimensional sequences.
>
> The model specification was also added. Encoder/Decoder have three layers RNN with LSTM neurons and the hidden dimension is 400.
>
> Nevertheless we have also repeated the same experiment with a different configuration to demonstrate the robustness of the proposed approach. The graphical visualisation is added to the main text as example two.
>
> Partial reconstruction of the original sequence is beneficial for two very simple reasons: (1) that it is an easier task; and (2) for large-scale industrial processes it is often very hard to diagnose where the problem comes from. A complete auto-encoder where the full input is reconstructed thru the RNN decoder would simply summarise the entire industrial process. Instead, partial reconstruction allows operators to focus on selected aspects of the process (e.g. pressure issues, temperature deviation... etc) and therefore have different clusters to reflect diagnostics of different aspects of the industrial system.

---

### Author Response · Authors · 2018-01-04
**Summary of updated paper (4-Jan-2018)**

Happy new year 2018 to everybody. We have updated the paper and here are the highlights:

-- Added detailed description of the problem including process graph in the appendix (Large compressor at a natural gas terminal)

-- Added dataset description and names of sensors used

-- Repeated experiement with a different configuration to further illustrate the idea of partial reconstruction and to ensure model robustness. Results are visualised graphically as the second example in the main text.

-- Referenced related works: both non-NN (DTW) and NN-based approaches. DTW works are dominated by unidimensional time series and wearable sensor where the dimensionality remain quite low. In NN world, related researches are dominated by well-labelled audio/video data where application use cases like ours is underrepresented. The closest one is about gesture recognition (recurrent autoencoder) but only very few sensors were involved.

-- We totally recognise that reconstructing a subset of the original sequence is indeed a much easier task. The key benefit is that it allows operators to diagnose different aspects of the machine rather than the machine as a whole. This is a practical benefit at the use case level.

-- Most related works have well defined dataset with bounded and labelled time series datasets. In this study we focused on an unbounded and unlabelled time series dataset which is admittedly more available in the real world. (Cost of collecting labelled time series is high, and more importantly it's an objective & manual exercise) By empirically demonstrating that unbounded/unlabelled time series data can be effectively summarised by vector representation in an online scenario and to apply clustering algorithms on them, it offers a useful practical tool for diagnostics and maintenance.

---

### Decision · Program_Chairs · 2018-01-29
**ICLR 2018 Conference Acceptance Decision**

**Decision:**

Reject

**Comment:**

This paper applies a form of recurrent autoencoder for a specific type of industrial sensor signal analysis.  The application is very narrow and the data set is proprietary.  The approach is not clearly described, but seems very straightforward and is not placed in context of prior work.  It is therefore not clear how to evaluate the contribution of the method.  The authors have revised the paper to include more details and prior work, but it still needs a lot more work on all of the above dimensions before it can make a significant contribution to the ICLR community.